# A Prospective Observational Cohort Study Comparing High-Complexity Against Conventional Pelvic Exenteration Surgery [note 1]

**DOI:** 10.3390/cancers17010111

**Published:** 2025-01-01

**Authors:** Charles T. West, Abhinav Tiwari, Yousif Salem, Michal Woyton, Natasha Alford, Shatabdi Roy, Samantha Russell, Ines S. Ribeiro, Julian Smith, Hideaki Yano, Keith Cooper, Malcolm A. West, Alex H. Mirnezami

**Affiliations:** 1Southampton Complex Cancer and Exenteration Team, University Hospital Southampton, Southampton SO16 6YD, UK; charles.west4@nhs.net (C.T.W.);; 2Academic Surgery, Cancer Sciences, University of Southampton, Tremona Road, Southampton SO16 6YD, UK; a.tiwari@soton.ac.uk (A.T.);; 3Finance Department, University Hospital Southampton, Southampton SO16 6YD, UK; 4Southampton Health Technology Assessments Centre, University of Southampton, Southampton SO16 6YD, UKk.cooper@soton.ac.uk (K.C.); 5Urology Department, University Hospital Southampton, Southampton SO16 6YD, UK; 6NIHR Southampton Biomedical Research Centre, Perioperative Medicine and Critical Care Theme, University Hospital Southampton, Tremona Road, Southampton, SO16 6YD, UK

**Keywords:** pelvic exenteration, cancer, surgical oncology, high-complexity pelvic exenteration, morbidity, health economics, quality of life

## Abstract

Advanced pelvic cancers, arising from organs such as the bowel, womb, and bladder, can be rapidly fatal without treatment. Pelvic exenteration (PE) describes an extreme operation that surgically removes multiple pelvic organs, and sometimes offers a potential cure, but involves a challenging recovery. Recent advances in surgery and patient care have facilitated even more extensive surgery to be conducted. These newer procedures are known as high-complexity PE and involve the removal of parts of the pelvic bones or large blood vessels. However, there is limited research in this field and concerns were raised about their impact on quality of life, cost-effectiveness, and increased complications. This study compares high-complexity PE to conventional PE, finding that the two procedures are closely similar in curing cancer, and provide acceptable quality of life, with minimal decision regret in patients. However, high-complexity PE consumes greater hospital resources partly due to a higher number of complications.

## 1. Introduction

Pelvic exenteration (PE) has established itself as the standard of care for patients with complex pelvic cancers [1]. The PelvEx Collaborative has provided a focal point of leadership, driving international collaborative research in the field, and highlighting improved surgical outcomes. Internationally more patients are now being considered for PE and previous anatomical contraindications to surgery are now challenged. This “higher and wider” approach encompasses routine resections of the pelvic girdle or major pelvic sidewall structures to achieve gold standard R0-resections [2].

PE developed in individual centres with heterogeneity in both patient selection and the operations performed, and historically this has resulted in inconsistent definitions within the literature. To more precisely describe the magnitude of surgery and improve data synthesis, the UK Pelvic Exenteration Network (UKPEN) constructed a now externally validated UKPEN lexicon [3,4]. In acknowledgement of the growing complexity of PE in contemporary practice, for the first time, this work defined a distinction between conventional PE and high-complexity PE: conventional PE, “A major surgical procedure where all or most organs in the [central] pelvic cavity are removed;” and high-complexity PE, “Encompassing conventional PE with the extension of surgery to remove bony structures or the structures within the pelvic sidewall. Pelvic sidewall structures include major vessels, sciatic nerves and/or bone. Bony structures include the sacrum and/or pubic bones” [3]. Examples of high-complexity PE are given in Figure 1.

The UK National Institute for Health and Care Excellence (NICE), who conduct health technology appraisals for the National Health Service (NHS), last reviewed the effectiveness of PE in 2020. NICE found a paucity of comparative and cost-effectiveness analyses, but recommended clinicians should consider referring patients with complex pelvic cancers to specialist PE units [5]. High-complexity PE is known to improve survival; however, there is the potential for higher incidences of post-operative complications that have healthcare cost implications [6]. To date, a cost analysis between conventional and high-complexity PE has not been conducted and the latter is not a commissioned service.

Whether conventional or high-complexity, PE remains a highly morbid procedure with the most recent PelvEx Collaborative results demonstrating a 63% post-operative complication rate [2]. These high levels of morbidity following PE make decision-making in these patients challenging, with a need to balance clinical, oncological, and health-related quality of life (HrQoL) outcomes rather than solely focusing on survival [2,7]. A further important outcome to consider is decision regret. This may be defined as a highly negative emotion and distress caused by the belief, after a critical decision, that the outcome would have been better had one selected a different choice, and is known to be associated with the quality of the shared decision-making process in cancer patients [8,9]. It becomes a critical outcome to measure when deciding whether to proceed with potentially morbid high-complexity PE. Nevertheless, there is little interest in the field of regret in the PE sphere to date [7]. Currently, there are no patient-reported outcome measures (PROMs) with adequate content validity for all patients undergoing PE, and in lieu of this, generic PROMs were used as pragmatic substitutes [4,10]. EQ5D-5L is the most commonly used generic PROM to assess HrQoL and has the additional benefit of being well established in cost–utility analysis [11].

When converting to Great Britain Pound (GBP) and adjusting for inflation by year of publication, PE is undeniably resource-intensive, with global perioperative costs conservatively estimated at GBP 52,436, rising to GBP 56,298 in a high-complexity unit [12,13]; therefore, identifying cost-savings is desirable. Cost–utility analyses are contingent on both survival and the levels of HrQoL experienced while alive, and, to date, while limited to only a few reports, survival rates of up to 15 years with acceptable HrQoL following PE are described [7,14]. In contrast, without surgery, median survival is typically only 12 to 18 months, with HrQoL deteriorating until death [15,16]. Only two cost–utility analyses were published in this field comparing no surgery to PE. Miller et al. 2000 reported an incremental cost-effectiveness ratio (ICER) of GBP 68,030 per quality-adjusted life year (QALY), while Koh et al. 2016 reported an ICER of GBP 89,904 per QALY [17,18]; although based on different health systems, both these ICERs far exceed NICE’s recommended willingness-to-pay threshold of GBP 30,000 per QALY [19]. These studies however likely underestimate the true cost-effectiveness of PE due to their limited time horizons, with PE having the potential to provide substantial survival and HrQoL benefits for a decade or more. When considering infralevator PE, additional costs from perineal reconstruction must also be considered. Myocutaneous flaps can increase operating time, incur plastic surgeon time, and prolong the length of stay [4,20]. Biological meshes offer a potentially simpler option, which, in systematic reviews, have not shown clinical inferiority to myocutaneous flaps [21,22]. Composite reconstructions using both myocutaneous flaps and biological meshes together were also reported for very substantial perineal defects [23,24]. To date, a large research gap exists as no studies have compared cost implications between conventional and high-complexity PE.

The Southampton Complex Cancer and Exenteration Team (SCCET) was established in 2009 following international recommendations to form advanced pelvic cancer multidisciplinary teams [25]. SCCET now serves a population catchment area of over five million, representing one of the largest experiences of PE in the UK. The present comparative observational cohort study aims to evaluate the performance of high-complexity PE for tumours that cannot be managed with conventional PE, against outcomes following PE for less aggressive disease not requiring extended pelvic resections. This paper reviews consequences that are important to clinicians, patients, and healthcare commissioners, with the objective of assessing survival, complications, HrQoL, and cost–utility; and includes a sub-analysis on perineal reconstruction cost-effectiveness to identify possible cost-saving opportunities.

## 2. Materials and Methods

The prospectively maintained consecutive SCCET database was evaluated (2010–2024) with the UKPEN lexicon applied to the database initially retrospectively using operation reports, histology, and post-operative imaging. SCCET patients are routinely offered at least yearly telephone follow-up indefinitely following PE as part of their ongoing assessment of HrQoL, imaging from referring hospitals is centrally available to SCCET, and referring institutions are routinely contacted by SCCET to minimise loss to follow-up or missing data bias.

PE was defined here as any resection involving two or more organs or compartments within the greater or lesser pelvis. Complexity of PE was defined as per UKPEN, with conventional PE encompassing central soft tissue resections, classified from the UKPEN lexicon as ≤P1, ≤A3, ≤C3, ≤SV0, ≤SN0, ≤PM1; and high-complexity PE as being more radical than this, with ≥P2, ≥A4, ≥SV1, ≥SN1, ≥PM2, E1, E2, E3, and E4. Coccygectomy alone and E5 resections including the concurrent removal of inguinal or hepatic oligometastic disease, appendicectomy, and resection of involved small bowel or colon pulled into a tumour mass were not taken into account when defining pelvic resection complexity [3]. If patients had previous surgery before PE, any historical conventional pelvic resections were added to the lexicon classification. Patients undergoing abdominal exenteration, palliative-intent PE, and historic PE without SCCET involvement were excluded. If patients had undergone redo-PE, they were not duplicated in the reporting of the results. The sample size was dictated by the number of eligible patients in the database.

Conventional and high-complexity PE were compared, and, for the purposes of a cost–utility analysis only, a third no-PE group was extrapolated from Koh et al., 2016 [17]. The primary outcome was overall survival; secondary outcomes were perioperative mortality, disease-free survival, local-disease-free survival, R0-resection rate, index admission major morbidity, overall major morbidity, resource use, and longitudinal HrQoL using the generic validated EQ5D-5L and decision regret scale PROMs [9,26]; the definitions of the outcomes are given in Appendix A. PROMs were consecutively collected at baseline pre-surgery and 3 months, 6 months, and 12 months post-operatively from December 2021. EQ5D-5L utility scores were calculated using the EuroQol Valuation Technology set for England, and Decision Regret Scores were derived as per the manual [26,27]; longitudinal scores were placed on a scatter plot and regression curves fitted to this. When reporting PROMs, to ensure dropout transparency, completion numbers at each time point were reported, with no imputation for missing data; however, when undertaking cost–utility analysis, as described below, missing EQ5D-5L data were handled in the same manner as Koh et al., 2016 [17].

Statistical analysis was performed in POSIT PBC RStudio (Version 2024.09.0+375) using a Shapiro–Wilks test for normality testing with a Mann–Whitney U test, and Fisher or Chi-squared tests for continuous and categorical data, respectively. Time-to-event analysis used Kaplan–Meier and log-rank tests. Median follow-up times are reported, and overall survival analysis utilised the NHS Summary Care Record database for mainland patients, linked to the primary care death certification NHS records. Overseas referring units were contacted during the survival analysis to minimise loss to follow-up. If loss to follow-up occurred, patients were censored from time-to-event analysis at the date of their last known contact. Missing data were assumed to be missing completely at random; therefore, a simple pairwise deletion strategy was otherwise used throughout all analyses.

Cost–utility analysis was conducted with the costs for individual patients using a bottom-up approach. High-value resource units used were collected for each patient, and unit costs were sourced from the 2023 NHS National Cost Collection, NHS Supply Chain, Drugs and Pharmaceutical electronic market information tool, British National Formulary, and the Personal Social Services Research Unit costs of health and social care [28,29,30,31,32]. Local hospital finance department micro-costing and consumable supplier quotes were used when reproducible alternatives were not available. Unit costs are summarised in Appendix A.

Two-state time-varying illustrative Markov models were constructed in Microsoft® Excel (Version 16.92) following best practice guidelines and the NICE reference case [33,34]. These models were used to estimate ICERs for conventional and high-complexity PE compared to no-PE. To estimate the long-term survival and health–utility benefits of PE, the no-PE comparator group was drawn from Koh et al. 2016, and extrapolated alongside the study groups over a 15-year time horizon with 3-month cycles and annual 3.5% discounting [17]. The assumptions used to derive transition probabilities, costs, and QALY accrual are given in Appendix A. EQ5D-5L utility scores were mapped to EQ5D-3L using the NICE-recommended crosswalk approach [35]. Models were scenario-tested using deterministic sensitivity analysis to disfavour the cost–utility of PE groups. Sub-group analysis was then completed as per the NICE technology evaluation manual for patients undergoing high-complexity components with vascular reconstruction, major resection of bone, and infralevator PE, with further assessment of different perineal reconstruction strategies [34]. Health economic analyses were conducted using means, Student *t*-tests, and ANOVA with Bonferroni corrections again in RStudio.

Ethical approval was granted by NHS North East—Newcastle and North 2 Research Ethics Committee (REC: 22/NE/0032), and the database registered on ClinicalTrials.gov (NCT05219058) as work package 1 of the Reconstruction in Extended MArgin Cancer Surgery study. STROBE and CONSORT-PRO guidelines were used to report the study; see Appendix A [36,37].

## 3. Results

There were 373 cases on the SCCET database, 27 patients underwent abdominal exenteration, 11 palliative-intent PE, 11 historical PE without SCCET involvement, and 5 patients had redo-PE with two database entries, leaving 319 patients proceeding into the analysis. Following the application of the UKPEN Lexicon, there were 64 (20.1%) conventional PE and 255 (79.9%) high-complexity PE. These groups were largely comparable; however, there was a higher proportion of gynaecological malignancies in the conventional PE group (*p* = 0.002) and more patients with metastatic disease in the high-complexity PE group (*p* = 0.03). Major resection of bone (*p* < 0.001), non-expendable vessel resection (*p* = 0.006), and composite perineal reconstructions (*p* = 0.05) were more frequent in the high-complexity PE group (details provided in Table 1).

There were no statistically significant differences in the primary outcome measure of overall survival, with a median survival of 10.5 years in the conventional PE group, and 9.8 years in the high-complexity PE group. The 5-year survival rates were 72.1% (CI 60.3–86.2) for conventional PE and 70.9% (CI 56.0–70.9) for high-complexity PE (*p* = 0.52; see Figure 2a). Perioperative mortality showed no differences (*p* = 1.00), with three deaths occurring within 90 days in the high-complexity group only; one was due to coronavirus, while the other two were related to muscle-invasive bladder cancer with early hepatic and peritoneal recurrences leading to liver failure and malignant bowel obstruction, respectively. Disease-free survival was lower in the high-complexity group. The 5-year disease-free survival rates were 65.0% (CI 52.5–80.5) for conventional PE and 47.2% (CI 40.3–55.4) for high-complexity PE (*p* = 0.034; see Figure 2b). Local-disease-free survival did not show significant differences, with local control rates at 5-years of 77.3% (CI 65.6–91.1) for conventional PE and 71.2% (CI 64.6–78.6) for high-complexity PE (*p* = 0.30; see Figure 2c).

R0-resections were achieved in 52/60 (87%) of conventional PE and 210/252 (83%) of high-complexity PE, with no significant difference (*p* = 0.08). A sub-analysis of R1-resections revealed the majority were continuous with the main tumour in both groups: 4/7 (57%) for conventional PE and 28/43 (65%) for high-complexity PE (*p* = 0.89). Major morbidity was significantly higher following high-complexity PE in both the index admission and until the date of the last follow-up, with 52/255 (20%) rising to 79/255 (31%), experiencing Clavien–Dindo complications ≥3a, with *p* = 0.02 in both comparisons; however, there were no significant differences in critical care lengths of stay (both 5 days, *p* = 0.85), or overall lengths of stay (18 days vs. 20 days, *p* = 0.56). Median follow-up was at least 29 months for both groups, without significant difference (*p* = 0.19). See Table 2 for further details.

In total, 60 patients, 10/60 (17%) conventional PE and 50/60 (83%) high-complexity PE, had prospective PROM data collected. No significant difference was noted between the two groups at 12 months post-operatively; however, significantly poorer EQ5D-5L mobility (*p* = 0.04) and pain (*p* = 0.049) scores were observed at 6 months in the high-complexity PE group, but these had resolved at 12 months. Decision regret scores were low overall in both groups throughout the 12 months, with no significant differences found between groups. Longitudinal PROM trends are given in Figure 3, and further detail on PROM analysis is available in Appendix A.

There were significant differences noted in healthcare resource usage between conventional and high-complexity PE, with mean costs being significantly higher in the high-complexity PE group (*p* < 0.001) by GBP 8462. The majority of this expense was found to be due to prolonged operating times, and higher numbers of subspecialty surgeons required for the high-complexity cases. The cost of increased incidence of major complications in high-complexity PE is also seen here with significantly higher expenditure on unplanned radiology, being GBP 65 higher (*p* = 0.02); see Table 3A. Infralevator PE was found to be the largest determinant of expense, even over the degree of complexity, with mean costs of GBP 22,199 more (*p* < 0.001) than supralevator PE; see Appendix A. Sub-group analysis of patients undergoing infralevator PE demonstrated reduced mean costs in the biological mesh group, with post hoc testing indicating biological mesh reconstruction was significantly lower than myocutaneous flaps by GBP 14,624 (*p* = 0.01), and composite reconstruction by GBP 24,880 (*p* < 0.001); however, composite reconstruction and myocutaneous flap reconstruction were not significantly different despite an increased mean cost of GBP 10,256 (*p* = 0.45); see Table 3B. Markov models estimated the cost per QALY to be GBP 7414 and GBP 10,077 for conventional and high-complexity PE, respectively. When this was modelled against the no-PE group extrapolated from Koh et al. 2016 [17], the ICERs were found to be GBP 2446 and GBP 5061, respectively; see Table 3C. Scenario-testing deterministic sensitivity analysis is presented in Appendix A; the maximum ICERs for conventional and high-complexity PE during this were GBP 6093 and GBP 9488, both with 50% decreases in costs of no-PE.

## 4. Discussion

This is the first study to formally categorise PE by complexity, finding comparable overall survival and local-disease-free control between conventional and high-complexity cases in one of the largest PE experiences in the UK. However, this benefit comes with a higher incidence of major complications, increased health resource use, and elevated costs. These findings may be of relevance to health technology appraisals of PE surgery. The two groups in this study were similar; however, there were more gynaecological tumours in the conventional PE group. Not unexpectedly from the study design, there were more major bone and non-expendable aorto-iliac axis resections in the high-complexity PE group.

Overall survival and local-disease-free survival plots appear remarkably similar for high-complexity vs. conventional PE; however, disease-free survival was slightly lower in the high-complexity group, as is consistent with an observed increase in systemic disease recurrences. This may be due to several factors, with patients requiring high-complexity PE by their very nature having anatomically less favourable tumours that may have more aggressive disease biology when compared to those in the conventional PE group; the tumour types in this study were not equivalent, potentially having varied responses to systemic chemotherapy, with less gynaecological cancers in the high-complexity group; the increased technical difficulty in obtaining an R0-resection imposed by the high-complexity PE may have contributed; and lastly, patients undergoing high-complexity PE will experience a higher magnitude of surgical trauma compared to conventional PE, and thus the consequent increased inflammatory response may somehow be contributing to increased systemic recurrence in a manner that is not currently well understood [38].

High-complexity PE appears relatively safe in the perioperative period with three 90-day mortalities—all of which were not directly attributable to surgery. R0-resection rates in both groups were favourable with 87% and 83% in conventional and high-complexity PE, respectively, with no significant differences between these. The non-significant, but slightly lower R0-rate in the high-complexity PE group may be due to either increased technical difficulties encountered when performing resections outside of the central pelvic compartment, or SCCET’s more pragmatic approach where intra-operative electron beam radiotherapy is applied to key margins at risk, particularly in certain high-stakes anatomical zones where further resection may substantially increase morbidity, such as the inner aspect of the acetabulum or over the sciatic nerve. These are preferentially treated with this modality as opposed to further surgical extension. This study also uses established UK colorectal cancer guidance to define an R1-resection [39]; however, it is acknowledged that this critical outcome is not yet universally agreed upon or measured consistently. This is recognised here with the reporting of continuous and discontinuous R1-resections given separately, with no significant differences found between conventional and high-complexity PE.

There was significantly higher major morbidity in the index admission following surgery in the high-complexity PE group. High-complexity PE encompasses more radical techniques in any of the pelvic compartments. The major morbidity observed in this study aligns with rates in the literature for specific pelvic compartments. Posteriorly, sacrectomy has an estimated 52% major morbidity rate; laterally, pelvic sidewall excision is associated with a 28% major morbidity rate; and anteriorly, pubic bone resections may result in overall complication rates of 70% [40,41]. The cumulative morbidity associated with multiple high-complexity excisions in a single patient remains unknown, although the overall magnitude of PE is thought to increase the risks of empty pelvis syndrome complications [4,42]. With increasing use, the UKPEN lexicon may be able to give additional insight into this in the future.

The PROM data reported here is largely not significant. When evaluating the trends in Figure 3 conventional PE mirrors trajectories previously reported, with decline and recovery over the 12 months [7]; however, in the high-complexity group, although the initial decline is similar, HrQoL recovery is less pronounced at 12 months. This may be due to a longer recovery time to return to baseline levels, or it may indicate that patients are less likely to fully regain baseline HrQoL following high-complexity PE. Regret describes a negative psychological and emotional state associated with feeling that one’s current situation would be preferable had a different path been chosen, with anticipatory regret being worry that taking the wrong course of action may lead to regret in the future. It is known that feelings of regret following cancer surgery can change over time [43], however, longitudinal decision regret scale data have never before been reported in the context of PE. Again, statistical differences were not found in this study, but trends indicate pre-surgery anticipatory regret is higher in patients with more complex disease; however, over the 12 months, decisional regret scores are the same despite the higher rates of morbidity and of distant disease recurrence in the high-complexity group. Regret is related to individual psychology and mental health, which is known to remain more stable following PE, with patients undergoing sacrectomy compared to PE with no sacrectomy, not having significantly different scores in this domain [7,44]. This may be a reflection of the recovery and adaptation process occurring following high-complexity PE as patients reach their post-operative “new normal” [45]. Despite the interest in shared-decision-making there is remarkably little data on decision regret following PE in this field [46]. Only Armbruster and colleagues have published using the Satisfaction with Decision Scale following PE for gynaecological malignancies, who similarly noted that despite reduced physical HrQoL, patients would make the same decision again [47]. From the point of view of counselling a patient considering high-complexity PE, it is therefore reassuring that despite the increased potential for major morbidity, decision regret scores remain low.

It is in the analysis of health resource use and costs where the differences between conventional and high-complexity PE are most stark, principally due to theatre costs from the prolonged operating times and increased numbers of subspecialty teams required, which were also reported elsewhere [13]. The high-complexity cases in this study incurred a significantly greater cost in the perioperative period, amounting to a mean increase in GBP 8462 per patient. This significant financial expense is justified on clinical grounds, as, without PE, the outlook for patients with advanced pelvic tumours is consistently and exceptionally poor, without any treatment median survival being 5–7 months [48]. Moreover, this short survival is accompanied by at times relentless pelvic pain refractory to standard analgesia due to the presence of an enlarging pelvic cancer in a confined bony cavity; malodorous discharge and sepsis; intractable bleeding; uncontrollable tenesmus; and social isolation. Radiotherapy alone or combined with chemotherapy may increase survival times by a few months more; however, this is not curative, as many patients may have had prior pelvic radiotherapy and thereby are unable to tolerate more, and relief of symptoms may occur only in a third of patients, with toxicity from treatment frequent, cumulative, and at times life-threatening. Consequently, death from these tumours is known for being one of the worst ways of suffering [48,49,50,51,52]. Compared to no surgery, this study demonstrates excellent survival for both conventional and high-complexity PE, with HrQoL and major morbidity that does not appear to induce significant decision regret at 12 months. In addition, following a non-surgery pathway also remains costly, with palliative surgery, oncological treatments, complications from disease progression, and best supportive care found to be on average GBP 35,072 or GBP 57,336 in different healthcare systems on converting to GBP and adjusting for inflation. Nevertheless, the ICERs for PE in the current literature are not in favour of NICE recommending any additional funding, as these are constrained by limited time horizons between approximately 26.2 and 42 months [17,18]. In lieu of a no-PE group in this study, an attempt was made to speculate on the potential cost–utility of high-complexity PE by extrapolating from a previously published Australian no-PE cohort [16,17]. Although just illustrative, ICERs of GBP 2446 and GBP 5061 for conventional and high-complexity PE, respectively, fall well below the NICE willingness-to-pay threshold even when scenario-tested. Consequently, PE may be regarded as a long-term investment in a patient’s future health, and when considering offering this procedure from an economical perspective, it may be especially beneficial for younger patients and those with a higher baseline HrQoL.

Although the cost–utility of PE compares well against the substantial costs of recently available molecular cancer therapies, it remains an expensive intervention [17]. Consequently, it is important to identify cost-reduction strategies that do not compromise surgical outcomes, such as Althumairi et al., 2016 who reported significant savings when PE is performed by high-volume surgeons and hospitals [53]. The sub-group analysis in this study demonstrates that infralevator PE, when requiring perineal reconstruction, consumes significantly higher healthcare resources. Furthermore, Appendix A indicates that infralevator resections are a stronger driver of overall cost than the level of complexity. When comparing perineal biological mesh to myocutaneous flap reconstruction, significantly reduced average perioperative costs were found, despite no significant differences in the numbers of flaps and meshes used in the conventional and high-complexity groups, with savings attributable to reduced operating times, reduced subspeciality surgeons, and reduced post-surgery lengths of stay. Although the cost-effectiveness of biological mesh has never been assessed in the context of PE, evidence from standard extralevator abdominoperineal excision has shown biological mesh implants are overall less expensive to use perioperatively than vertical rectus abdominis muscle (VRAM) flaps, despite the high costs of the implants themselves [54]. Conversely, in a large PE cohort, Risbey et al. 2024 reported VRAM flaps did not significantly increase costs; however, they did similarly identify major vascular and bone resections as adding significant expense [13].

Clinical performance of these methods of reconstruction in preventing empty pelvis syndrome complications was similar, with increased morbidity due to reconstruction itself seen in myocutaneous flaps; however, this is reported elsewhere and is the focus of a parallel paper [55]. The SCCET preference is therefore to offer patients biological mesh reconstruction if feasible, and where a good bulk of omentum is present, an omentoplasty aids pelvic filling and neo-vascularisation of the mesh. In cases where the perineal skin defect is anticipated to be larger, where vaginal reconstruction will be needed, or sacrectomy above the level of S4 is conducted, then a myocutaneous flap will be routinely offered. This study represents the largest series of biological collagen tissue matrix mesh used in PE, with the next a series of six [56]. Further clinical and economic evaluation of these implants over time is clearly required in future work.

### Limitations

Post-operative morbidity may be underestimated due to many patients living long distances from SCCET. It is therefore possible that despite best efforts of prospective capture and long follow-up times, the follow-up programme may not capture all complications and costs managed at local institutions with some, particularly if perceived as minor, not being reported. However, this would likely impact both groups similarly. Furthermore, the impact of oncological adjuncts was not included in the analysis. Hyperthermic intraperitoneal chemotherapy was only recently introduced and the number of these cases is small; intra-operative electron radiotherapy is more established but is currently under active investigation in a double-blinded randomised control trial.

A smaller number of high-complexity PE cases were much more costly. This sub-group is best represented by patients requiring composite biological mesh and flap reconstructions, all high-complexity cases. Many of these have so many components that they are performed over two days in theatre with the patient transferred to critical care overnight; they represent the most morbid and costly disease that SCCET feasibly currently treats. Although in this study there appears to be a clear delineation between conventional and high-complexity PE, this represents a limitation in the definitions generated from the UKPEN lexicon used here. This tool was developed as a clinimetric scale, highly effective at coding complex heterogenous clinical phenomena [57], which was translated to a set of ordinal scales for each component of PE. The UKPEN lexicon however was developed qualitatively and has limitations; it is currently unknown how the ordinal scales within different compartments interact with one another, despite this being the case for the majority of patients; how to code patients that had pelvic resections before PE, particularly for those having PE for recurrent disease; and what the impact of moving up or down a level in these scales is, in terms of surgical and economic outcomes [3]. The differences in costs for the composite reconstruction patients imply that a third “very” high-complexity PE group exists. The injury severity score is a well known clinimetric tool that was similarly historically developed qualitatively from physicians’ experiences and was refined to be able to predict patient outcomes, resource use, and costs [58,59]. The UKPEN lexicon hopefully represents an essential first step for the PE field that could become increasingly useful as its uptake increases. In future studies on PE that use the UKPEN lexicon, stratification could be considered, for example, reporting exclusively only on infralevator PE cases; alternatively, statistical methods, such as principal component analysis, have the potential to transform the clinimetric UKPEN lexicon into quantitative data that could then manage heterogeneity with techniques such as propensity scoring [60]. It should be noted that adjusting for any confounding variable was not attempted here, limiting the impact of this study. Of note, machine learning could also offer a solution, indeed with its use anticipated as part of the design of an international PE risk-prediction tool based on pre-operative data, which could also include a PE radiological roadmap constructed using the UKPEN lexicon [61,62].

Robustly designed disease-specific PROMs with adequate content validity are more sensitive to finding significant differences, as this study was largely only able to comment on HrQoL trends. No such universal tool exists for PE patients; however, the PelvEx Collaborative is supporting the development of such a PROM. This would have been superior to the generic PROMs used in this study, which fail to capture many issues specific to PE patients that would adversely affect their HrQoL [63,64]. PROM data in this study may not be missing at random, as patients with worse HrQoL and more regret may have been less likely to engage with research. Therefore, this study may have overestimated the positive HrQoL outcomes to those actually experienced. In addition, non-direct health effects such as financial toxicity and occupational impact were not included in the cost–utility analysis.

Local data for a non-exenteration cohort was not collected, with the population from Koh et al. 2016 used for the Markov modelling. This approach is flawed and can only be considered illustrative. Limitations of the previous study were extrapolated and further compounded by the different costs in an Australian health system, using QALYs derived from SF-6D questionnaires, and converting costs to pounds sterling using historical exchange and inflation rates [17]. Due to the reduced survival and detrimental HrQoL without PE, it is challenging, and potentially unethical, to derive a non-exenteration group. SCETT has a preference to offer PE when feasible, in the knowledge that costs will exceed remuneration. Despite these limitations, this is the first time cost–utility was mapped for PE over a more appropriate time horizon.

## 5. Conclusions

This study validates the UKPEN definition of high-complexity PE, demonstrating that these more radical procedures for more aggressive cancers appear to give overall survival and local disease control that is comparable to less advanced tumours undergoing conventional PE. This comes at a price of higher morbidity; however, similar decisional regret and HrQoL outcomes suggest that this cost is acceptable to patients, particularly when the alternative option without surgery is death within months, a declining HrQoL, and not insignificant health resource use required for palliative pathways.

While conducted in an established UK PE unit, the study may not be fully generalisable; however, it does also provide insight into the challenges faced by such institutions. Only 20% of cases were conventional PE, showing the potential unmet financial implications of more extensive surgery, but speculative modelling demonstrates that high-complexity PE is likely to be cost-effective. Clearly, a formal commissioning of this service will optimise the situation and facilitate the development of the infrastructure needed to better enhance patient care, enable further research, and improve training for the next generation of clinicians.

## Figures and Tables

**Figure 1 cancers-17-00111-f001:**
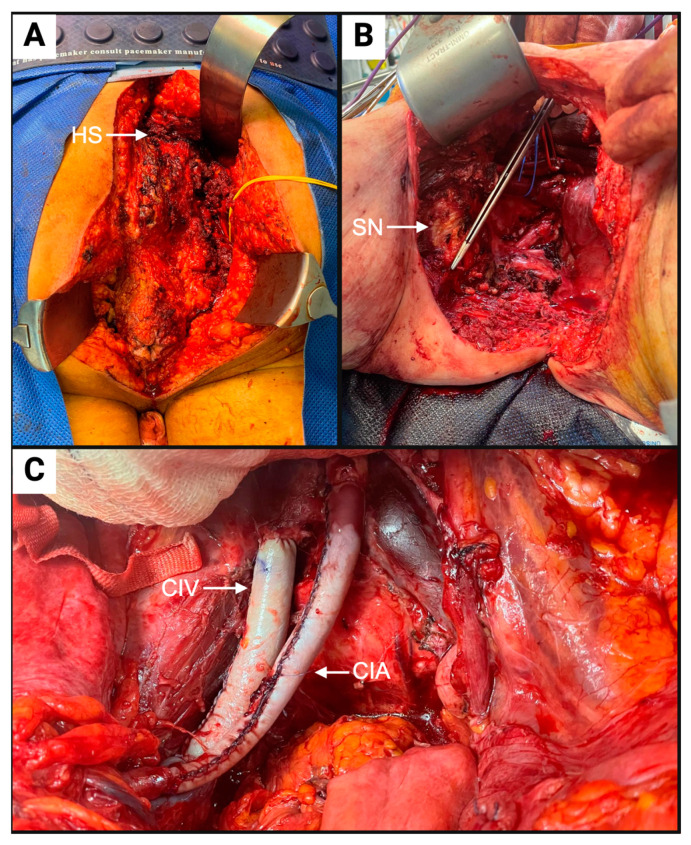
Intra-operative photos of high-complexity pelvic exenteration (PE): (**A**) Prone view of high-sacrectomy (HS) with arrow at level of division. (**B**) Complete sidewall resection exposing sciatic nerve (SN) with forceps indicating cut end of ischial spine. (**C**) Resection and reconstruction of common iliac artery (CIA) and common iliac vein (CIV) with bovine pericardium tube grafts.

**Figure 2 cancers-17-00111-f002:**
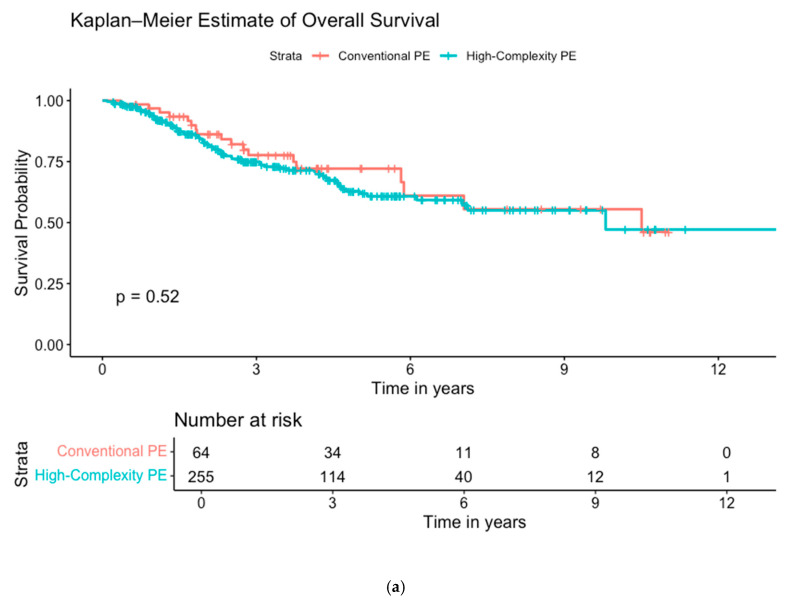
Kaplan–Meier survival plots, log-rank tests, and risk tables comparing conventional pelvic exenteration (PE) in red to high-complexity PE in blue: (**a**) overall survival, (**b**) disease-free survival, and (**c**) local-disease-free survival including only pelvic recurrences. Note: Five patients with benign disease were excluded from disease-free survival analyses.

**Figure 3 cancers-17-00111-f003:**
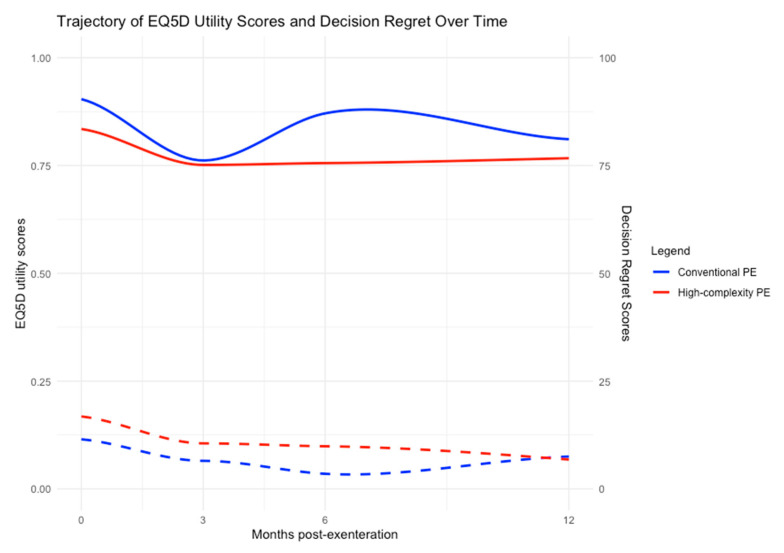
Trajectories of EQ5D-5L utility scores and decision regret scale (DRS) scores over time for conventional pelvic exenteration (PE) in blue against high-complexity PE in red; EQ5D-5L score lines are continuous and DRS scores are dashed. Individual points on scatterplot were removed and regression lines drawn with locally estimated scatterplot smoothing; left y-axis shows EQ5D-5L utility scores ranging from 0.00 to 1.00, with 0.00 representing death and 1.00 indicating excellent quality of life. Right y-axis shows DRS scores ranging from 0 to 100, with 0 representing no decisional regret, and 100 indicating maximal decisional regret. Dropout rates described in Appendix A.

**Table 1 cancers-17-00111-t001:** Baseline patient characteristics grouped by conventional pelvic exenteration (PE) and high-complexity PE. IQR = interquartile range; BMI = body mass index; COPD = chronic obstructive pulmonary disease; ASA = American Society of Anesthesiologists Score. Non-expendable vessels defined as aorta, common, or external iliac arteries or veins. * There were 4 smoking status and 12 ASA scores missing.

Baseline Patient Characteristics	Conventional PE	High-Complexity PE	*p*-Value
**Sample size,** *n (%)*	64 (20)	255 (80)	
**Sex,** *n (%)*			
Female	30 (47)	133 (52)	0.49
Male	34 (53)	122 (48)	
**Median age,** *years (IQR)*	66 (18)	61 (17)	0.08
**BMI**, *(IQR)*	26 (6)	26 (7)	0.95
**Smoking status *,** *n (%)*			
Current smoker	5 (8)	18 (7)	0.90
Ex-smoker	24 (38)	90 (36)	
Non-smoker	34 (54)	144 (57)	
**Comorbidities,** *n (%)*			
COPD/Asthma	8 (13)	27 (11)	0.66
Diabetes	5 (8)	24 (9)	0.81
Heart Disease	2 (3)	17 (7)	0.38
Chronic kidney disease	4 (6)	19 (8)	1.00
Hypertension	24 (38)	71 (28)	0.17
**ASA Grade *,** *n (%)*			
I	2 (3)	14 (6)	0.64
II	33 (55)	136 (55)	
III	25 (42)	93 (38)	
IV	0 (0)	4 (2)	
**Diagnosis,** *n (%)*			
Colorectal	45 (70)	188 (74)	0.002
Urological	3 (5)	38 (15)	
Gynaecological	10 (16)	13 (5)	
Anal	1 (2)	10 (4)	
Other	5 (8)	6 (2)	
**Cancer status,** *n (%)*			
Primary disease	40 (63)	155 (61)	0.56
Recurrent disease	21 (33)	99 (39)	
Metastatic disease	8 (13)	65 (26)	0.03
Neoadjuvant treatment	36 (56)	168 (66)	0.19
**Type of Exenteration,** *n (%)*			
Supralevator	38 (59)	155 (61)	0.89
Infralevator	26 (41)	100 (39)	
Perineal biological mesh reconstruction	22 (34)	61 (24)	0.60
Myocutaneous flap reconstruction	3 (5)	22 (9)	0.44
Composite mesh and flap reconstruction	0 (0)	15 (6)	0.05
Major bone resection	0 (0)	63 (25)	<0.001
Non-expendable vessel resection	0 (0)	24 (9)	0.006

**Table 2 cancers-17-00111-t002:** Secondary outcomes grouped by conventional pelvic exenteration (PE) and high-complexity PE. IQR = interquartile range. Note: Benign and disseminated peritoneal malignancy cases were removed from R0 analysis. * One R1 cases had both continuous and discontinuous margins of concern in single specimen.

Outcomes	Conventional-PE	High-Complexity PE	*p*-Value
**Sample size,** *n (%)*	64 (20)	255 (80)	
**Survival,** *n (%)*			
90-day mortality	0	3 (1)	1.00
**Margin status *,** *n (%)*			
R0	52 (87)	210 (83)	0.08
R1	7 (12)	42 (17)	
R2	1 (2)	0 (0)	
Continuous R1	4 (57)	28 * (65)	0.89
Discontinuous R1	3 (42)	15 * (35)	
**Morbidity,** *n (%)*			
Index admission major complications	5 (8)	52 (20)	0.02
Overall major complications	10 (16)	79 (31)	0.02
**Resource use**			
Median length of surgery, *minutes (IQR)*	557 (225)	678 (371)	<0.001
Median critical care use, *days (IQR)*	5 (4)	5 (3)	0.85
Median length of stay, *days (IQR)*	18 (16)	20 (17)	0.56
**Median follow-up time,** *months (IQR)*	34 (30)	29 (42)	0.19

**Table 3 cancers-17-00111-t003:** Health economic outcomes. (**A**) Cost-effectiveness comparing conventional pelvic exenteration (PE) to high-complexity PE, divided into overall costs and sub-categories of perioperative costs. (**B**) Cost-effectiveness sub-analysis of only infralevator cases using reconstruction with either biological mesh, myocutaneous flap, or composite (both myocutaneous and biological mesh). (**C**) Cost–utility analysis output from Markov model for conventional PE and high-complexity PE against extrapolated no-PE group. SD = standard deviation; QALY = quality-adjusted life-year; ICER = incremental cost-effectiveness ratio; NA = not applicable. Note: Five patients with incomplete data were excluded from analysis.

(A) Conventional vs. High-Complexity Cost-Effectiveness	Conventional-PE	High-Complexity PE		*p*-Value
**Sample size,** *n (%)*	63 (20)	251 (80)		
**Costs (GBP), *mean (SD)***				
Overall costs	37,271 (15,182)	45,733 (24,275)	<0.001	
Operating Theatres	18,875 (6988)	24,858 (11,326)	<0.001	
Admission	18,144 (9910)	20,454 (15,490)	0.15	
Unplanned radiology	146 (170)	211 (270)	0.02	
Re-interventions	107 (487)	210 (1028)	0.25	
**(B) Infralevator reconstruction cost-effectiveness sub-analysis**	**Biological mesh**	**Myocutaneous flap**	**Composite**	** *p* ** **-value**
**Sample size,** *n (%)*	80 (67)	25 (21)	15 (13)	
**Costs (GBP), *mean (SD)***				
Overall costs	51,996 (22,365)	66,620 (21,773)	76,876 (16,825)	<0.001
Operating theatres	26,873 (7277)	37,536 (14,131)	42,011 (7203)	<0.001
Admission	24,496 (17,707)	28,450 (11,758)	34,050 (11,811)	0.09
Unplanned radiology	275 (298)	269 (258)	346 (206)	0.64
Re-interventions	352 (1666)	365 (722)	470 (1037)	0.96
**(C) Cost–utility analysis**	**Conventional PE**	**High-complexity PE**	**No-PE**	
Total QALYs	6.04	5.23	1.44	
Total costs	44,750	52,677	33,516	
Cost per QALYs	7414	10,077	23,246	
ICER compared to no surgery	2446	5061	NA	

## Data Availability

The data, Markov model, and R code used in this study are available on request from the corresponding authors due to restrictions from a paper contributing to an ongoing PhD thesis, and on the ethically approved research protocol.

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
