# Peer review of "A Prospective Observational Cohort Study Comparing High-Complexity Against Conventional Pelvic Exenteration Surgeryâ€"

_cancers, 2025, doi:10.3390/cancers17010111_

Round 1
Reviewer 1 Report
Comments and Suggestions for Authors
The article is an observational cohort study evaluating the outcome of extended pelvic exenteration. The paper is formally very well structured, with a comprehensive and clear introduction, methods are reproducible, results meticulously reported and support the conclusions.
I am afraid about the comparability of the two techniques since, as acknowledged by the authors, the two cohorts differ in terms of the aggressiveness of the disease. This issue is quite transparent and deeply discussed. Anyway, at a superficial reading, the article may be considered a comparison of two options always available for each patient. Pelvic exenteration and extended pelvic exenteration have different indications. I suggest the authors stress this point to avoid the misinterpretation of trying to compare two very different cohorts. Otherwise, I would underline an interesting speculation that I am drawing after your research, that is the following: treating patients with extended pelvic exenteration allows for a disease-free survival that is comparable to that of less aggressive cases undergoing conventional pelvic exenteration at a reasonably higher cost and morbidity.
Author Response
Comments 1: I am afraid about the comparability of the two techniques, since as acknowledged by the authors, the two cohorts differ in terms of aggressiveness of the disease. This issue is quite transparent and deeply discussed. Anyway, at a superficial reading, the article may be considered a comparison of two options always available for each patient. Patient exenteration and extended pelvic exenteration have different indications. I suggest the authors stress this point to avoid the misinterpretation of trying to compare two very different cohorts.
Response 1: This has been made clearer in the sections of the manuscript most likely to be read: the abstract (page 1, paragraph 2, lines 34-36), aims (page 4, paragraph 1, lines 131-137), and conclusion
Comments 2: Otherwise, I would underline an interesting speculation that I am drawing after your research, that is the following: treating patients with extended pelvic exenteration allows for a disease-free survival that is comparable to that of less aggressive cases undergoing conventional pelvic exenteration at a reasonably higher cost and morbidity.
Comments 2: This point is made in the abstract and first sentences of the discussion, which we have not changed further. The conclusion has been re-written as per Reviewer 3 and we have emphasised this finding further here, by putting this observation as the first sentence in this section (page 15, paragraph 4, from line 518).
Reviewer 2 Report
Comments and Suggestions for Authors
As systemic treatments get better, the benefit of radical surgeries become more pronounced. We admire the authors dedication to both pushing the limit and considerations for QOL.
- In introduction there are multiple places where AUD is used and others where the $ is used. It is confusing to compare and say things are rising or falling when using different currencies. Recommend picking one as a standard and placing the rough conversion (with date of conversion) as a reference.
- DFS was lower in the high complexity group? It may be because gyn cancers are better responders to systemic chemotherapy?
- Why are R0 resections lower in the group getting a larger resection?
Great work
Author Response
Comments 1: In the introduction, there are multiple places where AUD is used and others where the $ is used. It is confusing to compare and say things are rising or falling when using different currencies. Recommended picking one as standard and placing the rough conversion (with date of conversion as reference).
Response 1: As the cost-utility analysis in this paper was conducted in 2023 GBP (£), we have used this to standardise all other values described in both the introduction and discussion to make this clearer. We have again used the same approach as described in Supplementary File 2 (Markov model parameters and assumptions) and added an additional note to this file. Explanations and revised costs now appear on page 3, paragraph 3, lines 104 – 114; and page 13, paragraph 2, lines 415 – 416.
Comments 2: DFS was lower in the high-complexity group? It may be because gyn cancers are better responders to systemic chemotherapy?
Response 2: The point on gynaecological cancers could partially explain the lower DFS, and we have added this to our discussion on why we found worse DFS in the high-complexity group (page 12, paragraph 2, lines 333 – 334). Due to the number of tumour types included in this study, we feel it would detract from the discussion to speculate further on chemosensitivity for each of these distinct pathologies.
Comments 3: Why are R0 resections lower in the group getting a larger resection?
Response 3: We believe this is due to the technical challenges of resections outside of the central pelvic compartment, and our preference for intra-operative electron radiotherapy in cases approaching borderline resectability R1 margins are more likely to be anticipated. We have revised this section (page 12, paragraph 3, lines 344 – 346).
Reviewer 3 Report
Comments and Suggestions for Authors
the manuscript is clear, relevant for this type of difficult Surgery of pelvic exenterations and presented in a well-structured manner with references of mostly recent publications and does not include a self-citation.
the study contains numerous tables and figures suggesting the importance of the study. the results are they easy to interpret and understand. pelvic exenteration surgery is a difficult surgery that is done in specialized centers, therefore I want to congratulate the authors for the results obtained. I just want the conclusions to be shorter and to the point. It does not reflect the importance and weight of the study. I would also like images with intraoperative aspects.
Author Response
Comments 1: I just want the conclusions to be shorter and to the point. It does not reflect the importance and weight of the study.
Response 1: The conclusions have been rewritten, also incorporating Reviewer 2’s point above (page 15, from paragraph 4, line 519).
Comments 2: I would also like images with intraoperative aspects.
Response 2: These have been added to the introduction to illustrate the potential difference in magnitude between high-complexity and conventional pelvic exenteration, with Figures appearing in the text thereafter renumbered as required (page 2, paragraph 2, from line 71).